# Monitoring of the Healthy Neonatal Transition Period with Serial Lung Ultrasound

**DOI:** 10.3390/children10081307

**Published:** 2023-07-29

**Authors:** Po-Chih Lin, Chia-Huei Chen, Jui-Hsing Chang, Chun-Chih Peng, Wai-Tim Jim, Chia-Ying Lin, Chyong-Hsin Hsu, Hung-Yang Chang

**Affiliations:** 1Department of Pediatrics, MacKay Children’s Hospital, Taipei 104217, Taiwan; ez8.4588@mmh.org.tw (P.-C.L.); waitim@mmh.org.tw (W.-T.J.);; 2Department of Medicine, MacKay Medical College, New Taipei City 25245, Taiwan

**Keywords:** infant, neonatal transition, lung ultrasound, lung liquid

## Abstract

Ultrasound has been used to observe lung aeration and fluid clearance during the neonatal transition period, but there is no consensus regarding the optimal timing of lung ultrasound. We aimed to monitor the trend of the serial lung ultrasound score (LUS) and extended LUS (eLUS) throughout the neonatal transition period (≤1, 2, 4, 8, 24, and 48 h after birth), assess any correlation to the clinical presentation (using the Silverman Andersen Respiratory Severity Score (RSS)), and determine the optimal time of the ultrasound. We found both LUS and eLUS decreased significantly after 2 h of life and had similar statistical differences among the serial time points. Although both scores had a positive, moderate correlation to the RSS overall (Pearson correlation 0.499 [*p* < 0.001] between LUS and RSS, 0.504 [*p* < 0.001] between eLUS and RSS), the correlation was poor within 1 h of life (Pearson correlation 0.15 [*p* = 0.389] between LUS and RSS, 0.099 [*p* = 0.573] between eLUS and RSS). For better clinical correlation, the first lung ultrasound for the neonate may be performed at 2 h of life. Further research is warranted to explore the clinical value and limitations of earlier (≤1 h of life) lung ultrasound examinations.

## 1. Introduction

The fetal lung is a fluid-filled organ, and the fluid is important for lung development in utero [1]. The fluid clearance from the lungs during the neonatal transition period is complex, poorly understood, and may start before, during, and after delivery [2,3,4,5]. The process includes mechanical force as the neonate passes through the birth canal and fluid resorption to the pulmonary epithelium, which is induced by increased adrenaline and vasopressin [6,7,8]. If the fluid clearance is inadequate, the neonate may develop transient tachypnea of the newborn (TTN) [9].

Lung ultrasound is now widely used to diagnose neonatal pulmonary diseases, such as respiratory distress syndrome, meconium aspiration syndrome, pneumothorax, and TTN [10,11,12,13]. The lung ultrasound score (LUS) proposed by Brat et al. [14], based on the presence of B-lines and/or consolidation of the anterolateral chest scan of six different regions, was applied to assess the need for mechanical ventilation, the necessity of surfactant replacement therapy for respiratory distress syndrome in preterm neonates, and prediction of bronchopulmonary dysplasia (BPD) [15]. There is another modified scoring system with additional posterior chest scans of the four regions: the so-called extended lung ultrasound score (eLUS) [13,16,17]. The lung is a stereo structure, and gravitational effects may impact the posterior lung more significantly than the anterior lung in some diseases, like neonatal ARDS and meconium aspiration syndrome [13,16,17,18]. Although related research is limited, eLUS has similar reliability for BPD prediction [18,19].

Lung ultrasound has also been used to observe the lung during the neonatal transition period [20,21,22], and changes from the initial “white-out” pattern to a complete aeration pattern have been documented. However, few studies have used LUS, which can evaluate the lungs semi-quantitatively. In addition, there is no consensus regarding the optimal timing for performing lung ultrasound during the neonatal transition period. The “white-out” pattern is assigned a higher score in the LUS system, but it can be part of the normal presentation of early neonatal transition rather than a pathological finding. Therefore, the timing of the lung ultrasound examination should be considered when interpreting lung ultrasound in the neonatal transition period.

This study used LUS and eLUS to evaluate neonatal lung transition in the first 2 days of life. We aimed to monitor the trend of the serial ultrasound score throughout the neonatal transition period, assess for any correlation to the clinical presentation, and identify the optimal time to perform lung ultrasound.

## 2. Materials and Methods

This was a prospective, observational study, and the participants were enrolled from a tertiary center in Taiwan between December 2019 and February 2022. In our routine practice, stable neonates are brought to the nursery for postnatal care. The duty doctor then performs a complete physical examination. If signs of respiratory distress worsen or persist beyond 24 h, the neonate is admitted to the newborn ward for further evaluation and treatment.

We enrolled neonates who were admitted to the nursery with a gestational age of ≥35 + 0 weeks. The enrolled neonates underwent serial lung ultrasounds. The exclusion criteria were (1) respiratory distress, i.e., the need for any additional respiratory support (oxygen hood, noninvasive ventilation, or even mechanical ventilation), (2) pneumothorax, (3) major malformations or chromosomal abnormalities, (4) failure to obtain consent from the parent, and (5) the inability to perform the first examination within the first hour of life.

Lung ultrasound was performed by a single experienced neonatologist who was well-trained in lung ultrasound to exclude potential interobserver variability. This operator must be on standby at the hospital for about 8 to 12 h in order to perform the first lung ultrasound in time (≤1 h of life). Furthermore, the operator must ensure that he can complete a series of lung ultrasound examinations over the next two days to be eligible for case enrollment. Images were acquired using a Philips CX-50 with an L15-7io broadband compact linear array transducer (7–15 MHz). The six scanning time points were within 1 h after birth and then at 2, 4, 8, 24, and 48 h of life. The neonates were placed on the radiant warmer table in the supine position for scans of six areas (upper anterior, lower anterior, and lateral sites of each lung) [14]. The neonates were then turned to the left lateral lying position to perform four scans of the back (upper posterior and lower posterior of each lung) [18]. The entire examination was performed as gently as possible to keep the neonate calm.

We employed a Picture Archiving and Communication System (PACS) for the storage of the images. These images were assessed with the lung ultrasound scoring system (LUS), which was scored using the number and crowdedness of B-lines and/or consolidation [14]. A score of 0 was defined as the presence of only the A-line or two or fewer B-lines. A score of 1 was defined as three or more B-lines. If there were crowded, coalescent B-lines and the depth of the subpleural consolidations was less than 1 cm, it was defined as a score of 2. A score of 3 was defined as extended consolidation with a depth of more than 1 cm.

The sum of the anterior six scores comprises the lung ultrasound score (LUS, ranging from 0 to 18) and the sum of the total ten scores of the extended lung ultrasound score (eLUS, ranging from 0 to 30) [18]. If the LUS or eLUS increased from one time point to the next, it was defined as “backsliding” [20].

Before performing the scan, the Silverman Andersen respiratory severity score (RSS), which is based on the presentation and patterns of upper chest movement, lower chest retraction, xiphoid retraction, nares dilatation, and expiratory grunt [23], and the examination time of the scan were recorded. Clinical data, such as the sex, gestational age, birth weight, method of delivery, APGAR score at 1 and 5 min, maternal medical condition (such as gestational diabetes, preeclampsia), and special prenatal medication (antenatal steroid or antibiotics) were collected from medical records.

Statistical analyses were conducted using SPSS v21.0 (IBM, Armonk, NY, USA). The LUS, eLUS, and RSS at each time point were compared using the paired sample *t*-test. *p* < 0.05 was considered statistically significant. Correlation analysis between the LUS and RSS and between the eLUS and RSS was performed by calculating Pearson’s correlation coefficient.

Serial RSS was compared using the independent sample *t*-test between the groups. *p* < 0.05 was considered statistically significant. The data were also stratified based on whether the neonate was born via cesarean section. Serial LUS, eLUS, and RSS were compared between the groups using an independent sample *t*-test. *p* < 0.05 was considered statistically significant.

## 3. Results

A total of 72 newborns were born during the period in which we could conduct complete examinations. Among them, four neonates were admitted to the neonatal intensive care unit due to prematurity and respiratory distress (between 26 and 31 weeks of gestation), and eight neonates were admitted to the newborn ward. Out of the latter, six were due to TTN, one was due to pneumothorax, and one was due to respiratory distress syndrome with a gestational age of 34 weeks. Additionally, 25 neonates were excluded from the study because they did not meet other criteria, with the dominant reason being the inability to obtain permission from the parents.

Finally, thirty-five healthy neonates were enrolled in the study. The mean gestational age was 38.3 ± 1.3 weeks, and two (5.7%) neonates were preterm. The mean birth weight was 2887 ± 359 g. Twenty-eight neonates were delivered vaginally and seven (20%) were delivered via cesarean section. Four neonates (11.4%) experienced prolonged rupture of the membrane (>18 h). The median APGAR scores were 9 and 10 at 1 and 5 min, respectively. The other characteristics of the neonates are shown in Table 1.

The serial LUS and eLUS over time and the mean scores of the serial scans are illustrated in Figure 1. The mean operation time of the ultrasound is shown in Appendix A. Both the LUS and eLUS had statistically significant differences for each scan (*p* < 0.05), except for three groups: between the first and second scans (≤1 h and 2 h of life, *p* = 0.12 and 0.081, based on LUS and eLUS, respectively), between the fourth and fifth scans (8 and 24 h of life, *p* = 0.254 and 0.065, based on LUS and eLUS, respectively), and between the fifth and sixth scans (24 and 48 h of life, *p* = 0.103 and 0.058, based on LUS and eLUS, respectively). There were statistically significant differences between the fourth and sixth scans (8 and 48 h of life, *p* = 0.044 and 0.002, based on LUS and eLUS, respectively) (Figure 2A,B). The detailed *p* and T values of the scores for each scan are demonstrated in Appendix B and Appendix C.

The median RSS of each exam was 3, 3, 2, 1, 0 and 0. Within 8 h, there were statistically significant differences for each scan (*p* < 0.05), except for one group: between the third and fourth scans (4 h and 8 h of life, *p* = 0.096). After 8 h, there were no statistically significant differences between the scans (Figure 2C). The detailed *p* and T values of the RSS for each scan are demonstrated in Appendix D. The Pearson correlation between the LUS at all time points and the RSS was 0.499 (*p* < 0.001), and the Pearson correlation between the eLUS at all time points and the RSS was 0.504 (*p* < 0.001). However, both the LUS and eLUS had low Pearson correlations with the RSS within 1 h of life (0.15 (*p* = 0.389) and 0.099 (*p* = 0.573), respectively). When the first time point was excluded, the Pearson correlation between the LUS and the RSS was 0.488 (*p* < 0.001), and the correlation between the eLUS and the RSS was 0.546 (*p* < 0.001).

According to the results from the LUS, a total of 19 neonates (54.3%) experienced backsliding, with one of them experiencing backsliding twice. This brings the overall number of backsliding cases based on the LUS to 20. In contrast, when considering the results from the eLUS, a total of 27 neonates (77.1%) demonstrated backsliding. Among them, 6 experienced backsliding twice, and 2 had backsliding three times, resulting in a total of 37 cases of backsliding based on the eLUS. The distribution time points of the backsliding and the score increments are presented in Table 2. There was no statistically significant difference in the serial RSS at each time point in the subgroups based on the presence of backsliding based on the LUS. If subgrouped by the presence of backsliding based on the eLUS, the only statistically significant difference of the RSS happened at the third scan (4 h of life, the mean RSS with no backsliding and with backsliding was 0.3 ± 0.7 and 0, respectively, *p* = 0.03).

When subgrouped by delivery method, gestational age, or preterm birth status (whether preterm or not), there was no statistically significant difference in serial LUS, eLUS, or RSS at each time point.

## 4. Discussion

This is a study employing both LUS and eLUS for the serial evaluation of neonatal transition within the first 48 h of life. Both scores decreased gradually and had similar statistical significance and differences among the serial time points. Although the gradual improvement in the lung ultrasound score suggested lung aeration and fluid clearance from the lung during the neonatal transition period [20,21,24], there was no statistical difference in the score within the first 2 h of life.

In animal models [25], after the first breath is taken, the lung aerates and lung fluid moves to the interstitial tissue, resulting in pulmonary edema. The extravascular lung fluid increases temporarily, and the clearance of the fluid occurs at 30–60 min of life rather than at birth. The interstitial tissue pressure increases in the first 4–6 h after birth and decreases with further clearance of interstitial fluid via the lymphatics and blood vessels [4,25,26,27]. In this study, there were no statistically significant differences between the results of the first scan (≤1 h of life) and the second scan (2 h of life) in the LUS and eLUS groups. Our finding is similar to that of Blank et al. [20], who used a different scoring system to evaluate the neonatal transition period and found no significant differences in the first four scans within 2 h of life (1–10 min, 11–20 min, 1 h, and 2 h). This concordant finding indicates that human neonates have a “plateau phase” of lung fluid clearance in the first 2 h of life.

Guo et al. [21] monitored the neonatal transition period with a different ultrasound scoring system and found no differences between 6 and 24 h. In our study, we had similar findings in that both the LUS and eLUS had no statistically significant differences between 8 and 24 h. Notably, both scores still showed statistically significant differences between 8 and 48 h of life. Our findings suggest that the neonatal transition period can initiate not only in the first few hours but also throughout the first 48 h of life.

We observed a higher severity of respiratory syndrome (RSS) within the first hour of life. The neonate’s breath improved gradually in the first 4 h of life and stabilized after 8 h. This finding is compatible with previous studies that showed gradual improvement in respiratory distress signs in the first few minutes to hours during the neonatal transition period [28,29,30,31]. In the current study, both the LUS and eLUS had a moderate positive correlation with the overall clinical presentation of RSS. A similar finding was reported by Loi et al. [18], who evaluated preterm (gestational age ≤ 30 weeks) neonates’ lungs upon admission to the NICU and found that both the LUS and eLUS had a moderate correlation to the RSS. In term and late preterm neonates with TTN, a significant correlation between the LUS and RSS has also been demonstrated [11,32].

Interestingly, despite the moderate correlation between the scoring systems and the RSS overall, the first scan (≤1 h of life, mean time of 37 ± 12 min) was the only time point with poor correlation between both the ultrasound score and RSS. There was still a significant correlation between both the LUS and eLUS with the RSS if the first scan was excluded. To our knowledge, in most studies focusing on the correlation between the lung ultrasound scoring system and the severity of respiratory distress, the scan time is not within 1 h of life. Loi et al. [18] described their scan on the first day of NICU admission and followed the protocol by Brat et al. [14], who described the mean scan timing as 2 h of life. In a study conducted by Li et al. [32], the average time for the initial lung ultrasound was over 2 h of life for both the TTN and healthy groups. The first scan was performed at 60–180 min of life in a study by Raimondi et al. [11], and they did not make comparisons among different time points.

In our study, the early exam time and use of the RSS as an indicator of respiratory distress may explain this discrepancy. The RSS comprises the degree of chest movement, chest wall retraction, nares dilatation, and grunting [10]. However, these signs of respiratory distress can be part of the normal neonatal transition period, facilitating lung aeration and fluid clearance and not always indicating respiratory morbidity [28,30,33]. Healthy neonates can have these signs in the first 30 min to 2 h of life [28,30]. Hein et al. [34] proposed a “rule of 2 h” to manage neonatal respiratory distress. They suggest a 2 h observation period to differentiate the normal neonatal transition from other diseases, and this practice may prevent the unnecessary referral of neonates.

The selection of healthy neonates in this study may have impacted our findings. The LUS can differentiate an ill neonate from a healthy neonate, according to Poerio et al. [35], who published an observational study among term and late preterm neonates born via cesarean section. They found that the LUS at 30 min of life was significantly higher in the ICU admission group. Of note, although the ICU admission group was more likely to have signs of respiratory distress than the non-ICU admission group, there was no statistically significant difference in the RSS between the two groups. Further studies should focus on the correlation between the lung ultrasound scoring system and additional parameters of respiratory distress in the first hour of life.

Blank et al. [20], using a different scoring system, reported that 47% of neonates in their study (≥35 weeks) had backsliding, and no backsliding occurred after 4 h of life. Our study showed a higher incidence of backsliding based on the LUS and eLUS (54.3% and 77.1%, respectively). Additionally, backsliding was still observed through 48 h of life. Despite these differences, the presence of backsliding provided little clinically significant value in this study. Neonates with backsliding had a lower RSS at just one time point, and the difference was small (as backsliding was based on the eLUS, the mean RSS with and without backsliding was 0 and 0.3 ± 0.7, respectively).

This study had several limitations. First, the major limitation is the small sample size. The enrollment of participants during the pandemic was difficult. With limited manpower and resources, conducting ultrasound examinations on time and frequently also resulted in difficulties in enrolling participants. Second, the selection of healthy neonates made it impossible to set a cutoff point for the lung ultrasound score to differentiate whether the infant was ill. Third, the ultrasound operator performed a physical examination of the RSS, which can provide a timely assessment of respiratory distress just before the scan but inevitably introduces operator bias. Future studies may encompass a larger number of cases and adopt a more comprehensive approach by including very premature or ill neonates, particularly those diagnosed with conditions such as TTN or RDS. The study’s strengths were that this was a prospective study, and multiple assessments were performed within 4 h of life, which is compatible with clinical practice. This research also provides information on the use of the eLUS, which is currently limited.

In conclusion, in healthy neonates, there is a “plateau phase” of lung fluid clearance during the first 2 h of life. Although both the LUS and eLUS have a moderate correlation with the RSS overall, there is a poor correlation within 1 h of life. We suggest the first lung ultrasound should be performed at 2 h of life. We also found that clearance is processed throughout the first 48 h of life. Further large-scale studies are required to confirm these findings.

## Figures and Tables

**Figure 1 children-10-01307-f001:**
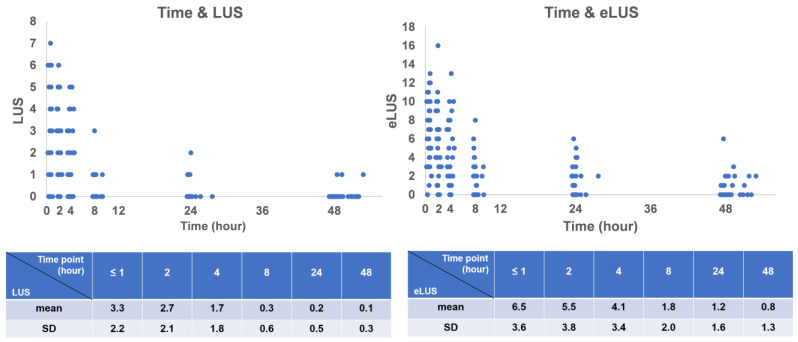
Serial LUS and eLUS by time. The upper scatter plot shows serial LUS and eLUS over time. LUS: lung ultrasound score; eLUS: extended lung ultrasound score.

**Figure 2 children-10-01307-f002:**
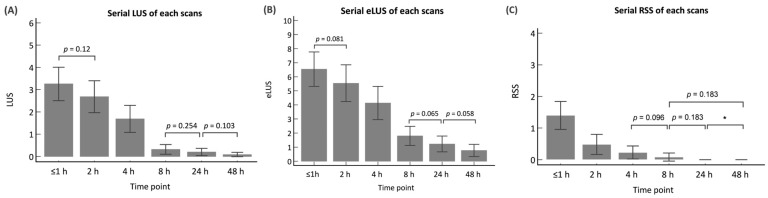
The serial changes in the LUS, eLUS, and RSS. The figure shows changes in the (**A**) LUS, (**B**) eLUS, and (**C**) RSS during the neonatal transition period. The *p* value demonstrated in the figure is the paucity without statistically significant differences at each time point. * RSS are all zero at both 24 and 48 h of life; therefore, there are no differences. LUS: lung ultrasound score; eLUS: extended lung ultrasound score; RSS: Silverman Andersen respiratory severity score.

**Table 1 children-10-01307-t001:** Characteristics of the neonates (*n* = 35).

Gestational age (weeks), mean ± SD	38.3 ± 1.3
Birth weight (g), mean ± SD	2887 ± 359
Term, No. (%)	33 (94.3)
Female/Male, No.	21/14
Cesarean delivery, No. (%)	7 (20)
APGAR score at 1 min, median (range)	9 (7–10)
APGAR score at 5 min, median (range)	10 (8–10)
Maternal GDM, No. (%)	2 (5.7)
Maternal preeclampsia, No. (%)	1 (2.9)
Antenatal antibiotics, No. (%)	9 (25.7)
Fetal distress, No. (%)	1 (2.9)
Prolonged rupture of membrane, No. (%)	4 (11.4)

None of the infants received antenatal steroids, precipitate labor, meconium-stained amniotic fluid, or resuscitation with positive pressure ventilation. GDM: gestational diabetes mellitus.

**Table 2 children-10-01307-t002:** Incidence of the backsliding and elevation of the score.

	Backsliding Based on LUS, *n* = 20		Backsliding Based on Extended LUS, *n* = 37	
Time Point	Incidence of Backsliding, *n* (%)	Elevation of the LUS, Mean ± SD (Range)	Incidence of Backsliding, *n* (%)	Elevation of the Extended LUS, Mean ± SD (Range)
2 h	9 (45)	2.2 ± 1.3 (1–5)	12 (32.4)	2.7 ± 1.8 (1–7)
4 h	5 (25)	1 ± 0 (1)	9 (24.3)	1.3 ± 0.5 (1–2)
8 h	2 (10)	1 ± 0 (1)	3 (8.1)	1.7 ± 0.6 (1–2)
24 h	3 (15)	1 ± 0 (1)	6 (16.2)	1.8 ± 1.2 (1–4)
48 h	1 (5)	1 ± 0 (1)	7 (18.9)	1.1 ± 0.4 (1–2)

## Data Availability

The datasets generated or analyzed during the current study are available from the corresponding author upon reasonable request.

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
