# Peer review of "Monitoring of the Healthy Neonatal Transition Period with Serial Lung Ultrasound"

_children, 2023, doi:10.3390/children10081307_

Round 1

Reviewer 1 Report

Although the sample is small, the study is well designed and interesting. In the future, I would suggest to include a larger sample to compare the results.

Author Response

Thanks for your kindly advices and I totally agree your words. The challenges posed by the pandemic era and the declining birth rate in Taiwan have presented difficulties for this study. However, despite these obstacles, we firmly believe that this research has significantly advanced our comprehension of the applications and limitations of lung ultrasound. Furthermore, we express our hope for future studies with larger sample sizes to validate our findings and optimize the utility of lung ultrasound in neonatal care. Thanks again for your comment, sincerely.

Reviewer 2 Report

Monitoring of the healthy neonatal transition period with serial lung ultrasound

by Po-Chih Lin, et al.

The paper shows descriptive data in the near term newborns, assessed by lung US, as standardization of the LUS and eLUS results. The correlation to RSS in the first hours and days of life is also made. The first hours/days are crucial to assess the neonatal transition interval and  the two first hours prompt to make decisions on the newborns affected  by RDS or not. It is worth noting that the improvement or impairment trend in the respiratory condition is allowing to pursuit the care as soon as possible. Consequently the objective evaluation by US may anticipate or confirm the diagnosis and the care, in the first days of life, allowing the potential respiratory support. (i.e. surfactant, CPAP, etc.).

Minor revision

Appendix A-C The line and columns should be clarified for audience by reporting the line and column headings and by avoiding to repeat “in compared with ..” each column.

Table 3 should be improved by avoiding to repeat headings and by enclosing them in legend of the table.

No further comments, please to refer to the above answers.

Author Response

1. Appendix A-C The line and columns should be clarified for audience by reporting the line and column headings and by avoiding to repeat “in compared with ..” each column.

We have changed the name of the appendix to “Comparative analysis of XX score at different time points” and delete the “in compared with” in the original column headings.( We have added an additional appendix, causing the original appendices A-C to become appendices B-D.) We hope that this modification will make it easier for readers to read and understand.

2. Table 3 should be improved by avoiding to repeat headings and by enclosing them in legend of the table.

We have change the abbreviation ”eLUS” to “extended LUS” for clarify. We hope that this modification will make it easier for readers to read and understand.

Thanks for your kindly advice and I have made modification according to your suggestion. Thanks again for your comment, sincerely

Reviewer 3 Report

This is a rather interesting study about the association between the lung ultrasound score and the respiratory severity score. My only major point is, however, is that I fear there is no practical applicability of this study in the clinical setting.

Minor points:

1.       Please, define severe respiratory distress, as it was used on row 69. Any kind of need for supplemental respiratory support was deemed severe RD?

2.       Please, state if prolonged rupture of membranes (row 112) was considered > 18 hours

3.       Please, provide better quality figures

4.       I think the information in Table 2 is unnecessary

5.       On row 136, please use median to express RSS, since it can only be given in full numerical values

6.       The information in rows 147-150 is somewhat difficult to read, please rephrase

7.       In the appendices, please used either “In comparison with…” or “Compared with…”

Again, my major fear is that the results from this study are somewhat useless, as they cannot be extrapolated to infants with any degree of RDS. Why would I be compelled to use LUS in healthy neonates, albeit in their transitional phase?

Author Response

1. Please, define severe respiratory distress, as it was used on row 69. Any kind of need for supplemental respiratory support was deemed severe RD?

Thank you for your kindly advice. We define any sick infant requiring additional respiratory support as having "severe respiratory distress."

We change the sentence to the following ”(1) severe respiratory distress, need for any additional respiratory support (oxygen hood, noninvasive ventilation or even mechanical ventilation)”

2. Please, state if prolonged rupture of membranes (row 112) was considered > 18 hours

We change the sentence to the following “Four neonates (11.4%) experienced prolonged rupture of membrane (>18 hours).”

3. Please, provide better quality figures

We have saved the image in higher resolution.

4. I think the information in Table 2 is unnecessary

We have relocated Table 2 to Appendix A. We have decided to retain the table as it highlights the findings within the first two hours of life. We believe it is still important to disclose the average operation time.

5. On row 136, please use median to express RSS, since it can only be given in full numerical values

We change the sentence to the following “The median RSS of each exam was 3, 3, 2, 1, 0 and 0

6. The information in rows 147-150 is somewhat difficult to read, please rephrase

We change the sentence to the following:

According to the results from lung ultrasound (LUS), a total of 19 neonates (54.3%) experienced backsliding, with one of them experiencing backsliding twice. This brings the overall number of backsliding cases based on LUS to 20. In contrast, when considering the results from extended lung ultrasound (eLUS), a total of 27 neonates (77.1%) demonstrated backsliding. Among them, 6 experienced backsliding twice, and 2 had backsliding three times, resulting in a total of 37 cases of backsliding based on eLUS.

7. In the appendices, please used either “In comparison with…” or “Compared with…”

We have changed the name of the appendix B-D to “Comparative analysis of XX score at different time points” and delete the “in compared with” in the original column headings. We hope that this modification will make it easier for readers to read and understand.

8. Again, my major fear is that the results from this study are somewhat useless, as they cannot be extrapolated to infants with any degree of RDS. Why would I be compelled to use LUS in healthy neonates, albeit in their transitional phase?

Thanks for your kindly advice and I have made modification according to your suggestion.

We conduct serial ultrasound scans throughout the neonatal transition period in order to establish the "normal range" of the lung ultrasound score within the first two days of life. This normal range holds potential significance for preterm neonates who are prone to lung diseases such as TTN or RDS. However, the frequent examinations and position changes required for scans can potentially violate the minimal handling protocol for the care of very preterm neonates. Therefore, in prioritizing safety, we have chosen to focus on late preterm and term infants for this study. The results of this study, which reveal a poor correlation between the ultrasound score and RSS within the first hour of life, highlights the need for further research to thoroughly examine the efficacy and limitations of early ultrasound. Besides, future studies may include a broader inclusion of very premature or sick neonates, specifically those with conditions such as TTN or RDS.

Thanks again for your comment, sincerely

Round 2

Reviewer 3 Report

I maintain my reserves regarding the clinical utility of this manuscript, but I have nothing to complain about in terms of scientific soundness. Larger and more diverse (in terms of respiratory distress) groups of subjects are needed in order to draw valid conclusions.

Author Response

I maintain my reserves regarding the clinical utility of this manuscript, but I have nothing to complain about in terms of scientific soundness. Larger and more diverse (in terms of respiratory distress) groups of subjects are needed in order to draw valid conclusions.

Ans:

Thanks for your kindly advices and I totally agree your words. Certainly, lung ultrasound or any other examination, like chest x-ray, provides limited clinical utility in managing "healthy" term or late preterm neonates. However, when signs of respiratory distress appear within 2 hours of life, potentially normal signs of neonatal transition can influence the assessment of whether the neonate is sick or not. The findings of this study emphasize the necessity for further exploration of the correlation between the ultrasound score and the degree of respiratory distress within 1 hour of life. This correlation becomes even more crucial when managing sick neonates using lung ultrasound as a diagnostic tool.

Thanks again for your comment, sincerely